# Learn Your Tokens:
# Word-Pooled Tokenization for Language Modeling

**Avijit Thawani**
thawani@usc.edu

**Saurabh Ghanekar**
USC

**Xiaoyuan Zhu**
USC

**Jay Pujara**
USC / ISI

## Abstract

Language models typically tokenize text into subwords, using a deterministic, hand-engineered heuristic of combining characters into longer surface-level strings such as 'ing' or whole words. Recent literature has repeatedly shown the limitations of such a tokenization strategy, particularly for documents not written in English and for representing numbers. On the other extreme, byte/character-level language models are much less restricted but suffer from increased sequence description lengths and a subsequent quadratic expansion in self-attention computation. Recent attempts to compress and limit these context lengths with fixed size convolutions is helpful but completely ignores the word boundary. This paper considers an alternative 'learn your tokens' scheme which utilizes the word boundary to pool bytes/characters into word representations, which are fed to the primary language model, before again decoding individual characters/bytes per word in parallel. We find that our moderately expressive and moderately fast end-to-end tokenizers outperform by over 300% both subwords and byte/character models over the intrinsic language modeling metric of next-word prediction across datasets. It particularly outshines on rare words, outperforming by a factor of 30! We extensively study the language modeling setup for all three categories of tokenizers and theoretically analyze how our end-to-end models can also be a strong trade-off in efficiency and robustness. [Github].

## 1 Introduction

Almost all natural language processing (NLP) begins with tokenization (Mielke et al., 2021). Sequences of characters are (mostly deterministically) segmented into discrete tokens, each of which has a lookup embedding in an enormous vocabulary matrix. Statistical NLP methods, similar to other forms of machine learning at the time, relied on feature extraction from these tokens, in the form of

| | Efficiency | Expressivity | Accuracy |
|---|---|---|---|
| Subword | High | Low | Mid |
| Byte/ Char | Low | High | Low |
| eByte/ eChar | Mid | Mid | High |

Table 1: Trade-offs involved when choosing tokenizers: Subword vs Bytes/Characters vs eByte/eChar (ours).

n-gram occurrences or part-of-speech tags or other representations of syntax. All of these pipelines have over time been replaced with end-to-end learning using recurrent neural networks (RNNs) or transformers, however the tokenization schemes remain static, deterministic, and manually engineered.

State-of-the-art approaches include subword tokenization schemes such as WordPiece (Wu et al., 2016), Byte Pair Encoding or BPE (Sennrich et al., 2016), and Unigram (Kudo, 2018), all of which are statistical methods for preprocessing a large unlabeled corpus of text to yield a fixed vocabulary, midway between characters or bytes at one end and whole words at the other. This results in a convenient trade-off in sequence description length while avoiding the UNK token, that is, a fallback mechanism for handling rare words. However, it is not obvious why these hand-engineered algorithms would be the optimal forms of tokenization and whether there exists a possibility for end-to-end models to also include this crucial stage of the NLP pipeline.

Recent work has shown countless limitations with subword embeddings. Several languages contain diverse morphological features whereas subword segmentation is mostly apt at only identifying suffixes and prefixes (Clark et al., 2022). Technical domains such as biomedical documents often need to pre-train their own tokenizer for improved vocabulary (Boecking et al., 2022). Finally, numbers are

often inconsistently segmented into subwords, leading to decreased arithmetic (Wallace et al., 2019) and estimation (Thawani et al., 2021b) skills. The extent of these numeric limitations is so dire that GPT-4 (OpenAI, 2023) has an explicit workaround of adding all numbers from 0 to 999 as individual tokens to the model's vocabulary.

Recently, several language models have been proposed which remove the tokenizer vocabulary entirely, beginning with a character (El Boukkouri et al., 2020) or byte-level (Xue et al., 2022) vocabulary and often compressing them into fixed units of around four tokens each (Tay et al., 2021; Yu et al., 2023; Clark et al., 2022). While these zero-assumption methods are useful in compressing text and consequently expand context windows, they completely ignore the word boundary. Besides, the so-called 'tokenizer-free' byte-based models are not entirely bias-free since the Unicode-8 encoding they use is itself biased towards representing Latin scripts with a single byte each, whereas some scripts[1] like Bammum (Africa), Meetei (India), and Cherokee (North America) may require four bytes to represent a single character.

The concept of words is a fundamental feature of nearly all human languages, including those written in Chinese or Japanese scripts that do not explicitly delineate words by whitespaces. This paper empirically studies the case where tokenizers lose their subword segmentation algorithms but utilize the word boundary for a multi-level model with added efficiency. More concretely, we use the word boundary to compress the base tokens of bytes or characters into word representations, which are then fed into the underlying language model (here, a small version of GPT (Radford et al., 2018)).

Our end-to-end learned tokenization undoubtedly has several limitations. It is not faster than subwords. It does not allow characters/bytes within one word to directly attend to those in another word. It relies on the word boundary, which is not straightforward to find for most internet-scale datasets. Nevertheless, we believe this empirical deep-dive into tokenizers for language modeling offers the following contributions:

1. We compare different tokenizer strategies for language modeling on multiple facets and on a fair footing across languages.

2. We are the first to explicitly use word bound-

[1] https://unicode.org/roadmaps/bmp/

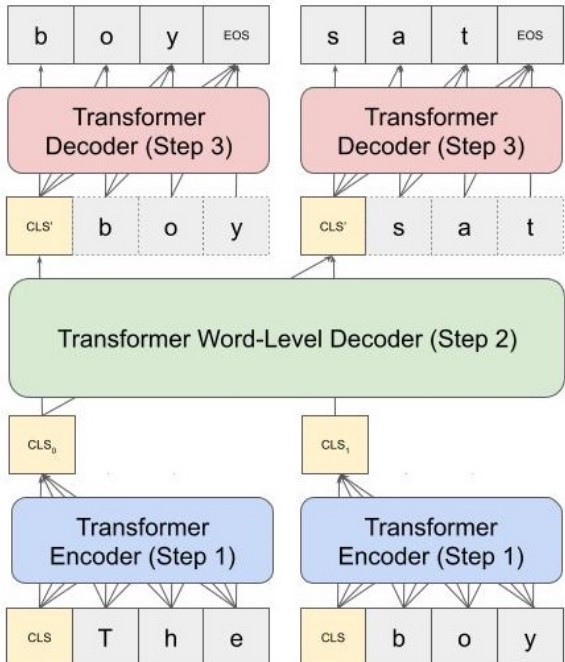

Figure 1: Overview of our proposed simple end-to-end tokenized autoregressive language model. A transformer encoder compresses the variable number of base units (here, characters) into n=1 CLS tokens per word. Dotted characters are the previously predicted tokens at inference, and when training they are the ground truth.

ary to compress an autoregressive language model's base tokens.

3. We report over 300% gains in language modeling capabilities over multiple languages and datasets, against both subwords and character/byte models, and by a factor of 30 on rare words.

4. We theoretically analyze strengths and weaknesses of our word-compressed tokenization scheme, which carries insights for the language modeling community.

We will publicly release all code (see supplementary material) and checkpoints upon acceptance.

## 2   Method

Figure 1 pictorially depicts our proposed language model architecture. Our end-to-end tokenization strategy is a straightforward word pooling method which uses a transformer encoder (Step 1) to pool in the base tokens (characters or bytes) into a fixed number of embeddings per word. This is analogous to how CLS embeddings are often used to pool in the embeddings of an entire sentence or any text

sequence in BERT-like transformer encoders. In our case, we have the equivalent of a fixed number of CLS tokens[2] prepended to each word that store the meaning of the entire word.

Next, (Step 2) the pooled per-word embeddings are passed onto the main language model, in our case, a vanilla transformer decoder like GPT (Radford et al., 2018). Finally, the contextualized word embeddings are fed through another transformer decoder to autoregressively decode the next word, one base unit (character or byte) at a time. Note that we call this method an end-to-end 'tokenizer' since it compresses the many units into a few embeddings per word, just like subwords, except the compression is learned from scratch. Finally, at decoding stage (Step 3), the contextualized word representations are unrolled with another transformer decoder to autoregressively predict one base token (character/byte) at a time. [3]

Note how we achieve our purported trade-off between subwords and byte/character models. The CLS representations learnt are unconstrained by a deterministic mapping as in subwords. They are also efficient to compute and decode from, since the first and last steps only allow intra-word attention. For a tokenizer-free model, roughly $80\%$ of the memory bottleneck[4] is spent on tokens from one word attending to tokens on another word, which we contest is of questionable importance relative to the overhead incurred.

Formally, we begin with a sequence of words $w_0, w_1, \ldots, w_n$ each of which is comprised of an ordered set of base units (character/bytes) $w_i = c_i^0, c_i^1, \ldots, c_i^{m_i}$ where $m_i + 1$ is the length of the $i$th word. The task is autoregressive language modeling, i.e. given the previously seen words $w_0, w_1, \ldots, w_{i-1}$ as well as the previously seen units in $w_i$ (the current word): $c_i^0, c_i^1, \ldots, c_i^{j-1}$ predict the next unit $c_i^j$.

**Character/byte** level models ignore the word-boundary and directly model the task as:

$$c_i^j = Decoder(c_0^0, \ldots, c_0^{m_0}, c_1^0, \ldots, c_i^0, \ldots c_i^{j-1})$$

**Subword** segmentation maps the base units deterministically into fewer subwords per word, i.e. , $w_i = c_i^0 \ldots c_i^{m_i} \to s_i^0 \ldots s_i^{m_i'}$ where $m_i' \leq m_i$, the number of subwords that the $i$th word is decomposed into. Following this determinsitc process, a subword model predicts the next subword as:

$$s_i^j = Decoder(s_0^0, \ldots, s_0^{m_0'}, s_1^0, \ldots, s_i^0, \ldots s_i^{j-1})$$

**Our end-to-end models** instead follow a three-step process to (1) pool base units into a fixed set of embeddings per word, (2) autoregressively predicting the next word embedding, and (3) autoregressively predicting individual unit embeddings per word:

$$CLS_i = Encoder(c_i^0, c_i^1, \ldots, c_i^{m_i}) \quad (1)$$

$$CLS_i' = Decoder(CLS_0, CLS_1, \ldots, CLS_{i-1}) \quad (2)$$

$$c_i^j = Decoder(CLS_i' \bigoplus c_i^0, \ldots, c_i^{j-1}) \quad (3)$$

Here, *Encoder* refers to a transformer BERT-like encoder and *Decoder* refers to a transformer GPT-like decoder. From an implementation standpoint, we prefix a fixed number ($n = 1$ or $4$ in this paper) of CLS tokens to every word before passing it through a transformer encoder. The word-level contextualized representations obtained on the other end are collectively depicted here as $w_i$.

Figure 2 is a visualization of how our end-to-end model saves on self-attention computation bottleneck by only allowing intra-word attention at the first step, before allowing contextualization of information across the word boundary in step 2 using the base decoder model. Finally step 3 again restricts the individual characters/bytes to be predicted using only the single word-level predicted embeddings.[5]

## 3 Experiments

There are numerous NLP tasks that can benefit from improved tokenization, such as Machine Translation, Question Answering, and Text Classification. However, the scope of our preliminary analysis is not to cast a wide net over every downstream application. Instead, we choose to analyze in depth the most commonly used pre-training task in NLP i.e. language modeling.

We pretrain autoregressive language models from scratch using different tokenizers described in

---

[2]This paper uses 4 CLS tokens per word, except in Section 4.2 where we ablate with 1 CLS per word.

[3]This autoregressive decoder can also be replaced by a non-autoregressive transformer which emits the entire word in $\mathcal{O}(1)$ time. Our initial experiments with such a vanilla setup performed much worse than autoregressive models (in line with prior work), therefore we leave this to future work.

[4]In Figure 2, this is the difference in blue attention blocks depicted in Figure between Byte/Char-level models and our intra-word attention.

[5]Note that our current implementation has minor deviations from the shown simplistic figure. Refer to Section 3.4 for details.

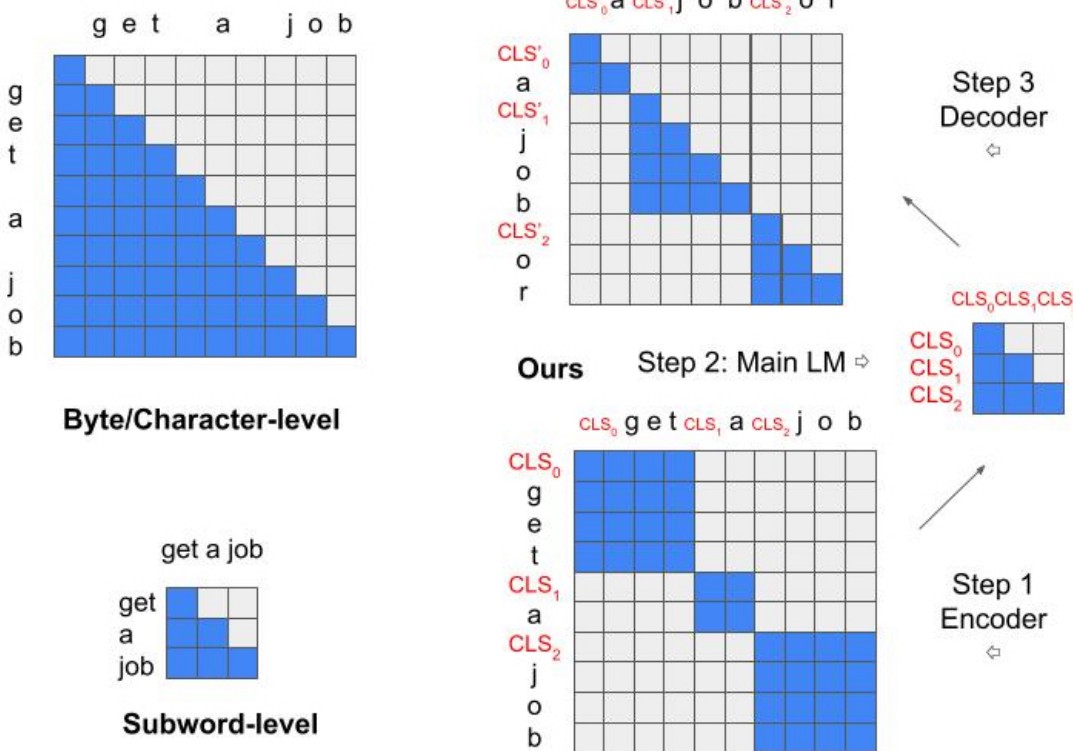

Figure 2: Self-attention visualized across (1) Byte/Char-level models, (2) Subword/Word-level models, and (3) Our proposed end-to-end tokenization modules (word encoder; base LM decoder; word decoder) with character base. Blue blocks indicate self attention mask. @ symbol indicates a prepended CLS token per word.

the previous section, on different datasets described in Section 3.2.

## 3.1 Models

We report results over the following tokenizers: **Subword**: a pretrained BPE vocabulary used by GPT-2 and GPT-3. **Byte**: a pretrained byte-level vocabulary as implemented in ByT5 Xue et al. (2022). **Character**: a corpus-specific vocabulary learnt from each dataset, with a fallback to UNK for characters unseen in training. **eByte/eChar**: Our end-to-end tokenized models which begin with the above Byte/Character vocabularies, but are compressed into CLS representations as described in Section 2.

There can be countless ways to make for a 'fair' comparison across tokenizers. We train all models on all datasets for the same number of total epochs. We also focus on letting the models access the same context window size, i.e. amount of information available to predict the next set of tokens. Different tokenizers can use vastly different memory sizes to fit the same amount of information. This is analogous to how the same book can be published

| Dataset | Size (MBs) | Words (Mil.) | Chars/ Word |
|---|---|---|---|
| English | 4.7 | 1.34 | 5.46 |
| French | 5.1 | 1.55 | 5.18 |
| Russian | 7.5 | 1.18 | 6.39 |
| Numeracy | 6.6 | 1.35 | 5.09 |

Table 2: Statistics for our language modeling datasets. See Section 3.2 for more details.

in different font sizes to choose between light and bulky books. We control for this information parity by fixing the number of characters in the available context to 192 for each tokenizer and each dataset. Subword models will then be allowed to access 192//N subwords where N is the average number of characters per subword.

## 3.2 Datasets

Our proposed method requires access to a word boundary signal, which can either be obtained from a clean natural language corpus, or by running a preprocessing pipeline on an unclean corpus to fil-

| Tokenizer | Acc (%) | Mem (GBs) | Params (Mil.) | Acc (%) | Mem (GBs) | Params (Mil.) | Acc (%) | Mem (GBs) | Params (Mil.) |
|---|---|---|---|---|---|---|---|---|---|
| Language | | English | | | French | | | Russian | |
| Subword | 14.37 | 0.55 | 76.8 | 41.20 | 1.50 | 76.8 | 8.31 | 1.49 | 76.8 |
| Byte | 13.69 | 0.53 | 25.7 | 17.39 | 0.54 | 25.7 | 12.76 | 0.53 | 25.7 |
| Char | 13.68 | 0.54 | 26.3 | 16.95 | 0.53 | 25.7 | 10.01 | 0.54 | 26.1 |
| eByte | **44.17** | 3.84 | 38.7 | 46.44 | 6.01 | 38.7 | 35.00 | 4.92 | 38.7 |
| eChar | 42.94 | 2.94 | 39.2 | **47.06** | 3.59 | 38.7 | **37.15** | 3.95 | 39.0 |

Table 3: Word Prediction Accuracies (Acc %) for different languages and tokenizers. See Section 4.1 for details.

ter out nonlinguistic tokens such as URLs or metadata. We chose the former to avoid confounding our results with a layer of preprocessing decisions. Therefore, our datasets are smaller but cleaner than the large-scale mC4 and OSCAR datasets typically used for training large language models.

Our choice of languages depended on the availability of a large enough corpus of clean data. We also deliberately avoid Chinese and Japanese corpora since segmenting them into words would require an additional, possibly confounding step of segmentation through an off-the-shelf model.

Concretely, here are the four datasets we pretrain and evaluate our language models on:

1. **English**: We randomly sample 10,000 paragraphs from the comprehensions of SQuAD2.0 (Rajpurkar et al., 2016) dataset.

2. **French**: We randomly sample 10,000 paragraphs from the comprehensions of SQuAD_FR (Cattan et al., 2021) dataset.

3. **Russian**: We randomly sample 10,000 paragraphs from the reading passages of the SberQuAD (Efimov et al., 2020) dataset.

4. **Numeracy**: We sample 60,000 rows of number-annotated sentences from WikiConvert (Thawani et al., 2021a), itself derived from the English Wikipedia. The task is to estimate these numbers approximately using the preceding words as context.

Table 2 presents statistics for the datasets that we use. The average dataset consists of 7.4M characters (676 unique) and 1.4M words (102k unique).

### 3.3 Metrics

Since the models have different vocabularies, we can not compare their perplexity scores. Instead, we fix the number of context to be exactly 192 characters and report the accuracy of predicting the next word (over held-out validation data from the same corpus as the training set). When estimating numbers, we report magnitude-based metrics that are typically reported in the literature (Thawani et al., 2021a; Berg-Kirkpatrick and Spokoyny, 2020): the order-of-magnitude Exponent Accuracy (EAcc: whether the number of digits are the same in the ground truth and predicted number) and Median Absolute Percentage Error (MdAPE: median of $100|x - y|/y$ where x is the prediction and y is the ground truth number).

### 3.4 Implementation

Every model (using a different tokenizer) is pretrained from scratch on every dataset described above. We report the aforementioned metrics on the individual test set from each corpus. Our base language model is a decoder-only transformer called minGPT[6] with 8 layers. For our end-to-end models, the main language model (Step 2) remains the same - with 8 layers like the others, whereas the word-encoder (Step 1) and word-decoder (Step 3) are both shallow transformers (encoder and decoder respectively) with 2 layers each. They use padding tokens to make each word of equal length for ease of training. We use trained absolute positional embeddings for all models, and the end-to-end models use it thrice - one for each step. We pretrain all models on all datasets from scratch for 100 epochs.

We set the learning rate to 0.0001, batch size to 2, and block size to 192. We used AdamW as our optimizer and trained our models on NVIDIA A100-PCIe-40GB GPUs. With this configuration, training each model variant for 100 epochs took an average of 52 hours.

---

[6] https://github.com/karpathy/minGPT

| Lang | Tok | 1 CLS | 4 CLS | Δ% | Δ Mem |
|---|---|---|---|---|---|
| en | eByte | 31 | **44** | 42% | 0.02 |
| en | eChar | 29 | **43** | 48% | 0.02 |
| fr | eByte | 31 | **46** | 48% | 0.02 |
| fr | eChar | 34 | **47** | 38% | 0.02 |
| ru | eByte | 26 | **35** | 35% | 0.02 |
| ru | eChar | 29 | **37** | 28% | 0.02 |

Table 4: Word Prediction Accuracies for different representative power (number of prefix CLS tokens) per word in our end-to-end byte/char-tokenized (Tok) models. Up to 45% higher prediction scores are available for a marginal increase in memory (Mem in GBs) of about 20 MBs. See Section 4.2 for details.

## 4 Results

### 4.1 Main Results

Our main results are summarized in Table 3. Next word prediction accuracies over different datasets show that given a fixed context window, our end-to-end tokenized language models perform much better (up to 300% from 14% to 44% on English) on all datasets than both the default BPE subwords as well as the tokenizer-free character and byte models. This does come at a doubling of GPU memory requirements, due to the additional word-level modules in our architecture.

### 4.2 Representation Power

Here we ablate the representative power available for word-pooling of character- or byte-level embeddings. This hyperparameter is controlled simply by adding a different (yet fixed) number of prefix CLS tokens per word before encoding via a transformer. Table 4 shows the word prediction accuracies and relative jumps when the number of prefix CLS tokens per word is increased from 1 to 4. We notice a huge jump for every model, with the trade-off in sequence description length. Note, however, that the memory usage does not jump by more than 20 MBs. Similarly, the number of parameters also increases (not shown in table) by only 300K (0.7%) for both eByte and eChar models.

### 4.3 Predicting Rare Words

One of the primary motivations for subword tokenization is their ability to compositionally create rarer words using other frequently occurring subwords. Wolleb et al. (2023) recently show that such compositionality is a significant contribution to the

| Tokenizer | Rare | Frequent |
|---|---|---|
| Subword | 0.11 | 7.20 |
| Byte | 0.00 | 4.36 |
| Char | 0.28 | 9.84 |
| eByte | 5.90 | 42.90 |
| eChar | **6.78** | **44.17** |

Table 5: Case study: Word Prediction Accuracies for Russian across tokenizers, stratified by Rare and Frequent words. See Section 4.3 for details.

| Tokenizer | % Num ↑ | EAcc ↑ | MdAPE ↓ |
|---|---|---|---|
| Subword | 20.0 | 44.8 | 95.72 |
| Byte | 39.9 | 40.5 | 99.00 |
| Char | 42.8 | 46.6 | 92.5 |
| eByte | **47.5** | **49.9** | **88.37** |
| eChar | 46.7 | 45.6 | 90.0 |

Table 6: Number Estimation results on Numeracy dataset across tokenizers. % Num = the percentage of times the model predicts a number, over which the next two metrics are calculated. EAcc = Exponent Accuracy. MdAPE = Median Absolute Percentage Error. See Section 4.4 for details.

empirical performance gains achieved by subword models. Hence, we report in Table 5 the word prediction accuracies for rare words (those seen less than 10 times in the training dataset) as well as frequent ones (those seen more than 45 times). We find our end-to-end models outperform by a factor of 5-7 on frequent words and over 30 times on rare words!

### 4.4 Number Estimation

We further evaluate a representative subset of tokenizers on WikiConvert number estimation task. Table 6 again reports that the ability of end-to-end-tokenized eByte/eChar is far better than both subword as well as Byte/Char models.

[]

## 5 Efficiency Analysis

Here, we determine the theoretical training and inference/generation speed-up accessible by compressing words using our end-to-end tokenizer as opposed to tokenizer-free methods, while also comparing against the more efficient subword models.

| Dataset | En | Fr | Ru | Ru (rare) | Ru (freq) | Numeracy [3] | |
|---|---|---|---|---|---|---|---|
| Metric | | Next Word Prediction Accuracy | | | | EAcc↑ | MdAPE↓ |
| **Subword** | 14.37 | 41.20 | 8.31 | 0.11 | 7.20 | 44.8 | 95.7 |
| **Byte** | 13.69 | 17.39 | 12.76 | 0.00 | 4.36 | 40.5 | 99.0 |
| **Char** | 13.68 | 16.95 | 10.01 | 0.28 | 9.84 | 46.6 | 92.5 |
| **eByte** | **44.17** | 46.44 | 35.00 | 5.90 | 42.90 | **49.9** | **88.4** |
| **eChar** | 42.94 | **47.06** | **37.15** | **6.78** | **44.17** | 45.6 | 90.0 |

## 5.1 Training Speed-up

Assume M total memory budget available (say, in GBs) and a context window of $T$ characters per batch. Also assume the tokenizer-free model (henceforth referred to as base/baseline) is a decoder transformer with $L$ layers and $D$ dimensions. The memory footprint would most significantly depend on the number of activations stored [7] which can be estimated as $M = LDBT^2$ where $B$ is the batch size. Given a fixed memory budget of $M$ and required context size of $T$ characters, we can find our optimal batch size as:

$$B = \frac{M}{LDT^2}$$

Assuming the training corpus comprises of $N$ characters, the number of training iterations required is:

$$X = \frac{N}{BT} = \frac{NDLT}{M}$$

Next, for subwords, a similar batch size can be estimated as:

$$B' = \frac{M}{LDT^2/s^2}$$

where $s$ is the number of characters per subword (roughly 2.8 for our three languages). Substituting to find the number of training steps:

$$X' = \frac{N}{B'T} = \frac{NDLT}{Ms^2}$$

The training speed-up of a subword model is therefore estimated to be $X/X' = s^2 = 7.8$x.

Finally, we calculate the analogous number of training steps required for one epoch of our end-to-end character-tokenized model. We assume $L/4$ word-encoder layers, $L$ primary LM (word level) layers, and $L/4$ word-decoder layers for simplicity

(this is our default setup in this paper). Let $B''$ be the optimal batch size that we wish to calculate and $c$ be the average number of characters per word (roughly 5.5 for English). Note that we retain $T$ characters as our context window, therefore the average number of words per batch sequence will be $T/c$. The memory footprint of activations would then be $(LDB''Tc)/4$ for the word encoder (and same for word decoder) and $(LDB''T^2)/(c^2)$ for the primary (word level) language model.

This leads to the optimal batch size:

$$B'' = \frac{M}{LDT(c/2 + T/c^2)}$$

and the number of training steps to be:

$$X'' = \frac{N}{B''T} = \frac{NDL}{M}(c/2 + T/c^2)$$

Finally, we estimate our proposed speedup in total training time as

$$X/X'' = \frac{T}{c/2 + T/c^2}$$

Plugging in $c = 5.5$ as a conservative number of characters per word[8] and $T = 192$ context window length, we get a $6.8x$ speed-up in training steps, which is only marginally less than the subword speed-up (7.8x) relative to character level langauge model.

## 5.2 Generation Speed-up

Another unique advantage of our end-to-end tokenized model is in generation, which is also parallelized per word. A character/byte model must generate one token at a time, then feed the predicted token back into the input and run the forward loop again for autoregressively generating the next. Assuming the $L$ layers of a decoder take $t$ seconds for the forward iteration of GPT-like decoder, the generation speed for such a character based model will be $1/t$ characters per second.

---

[7]The other components such as parameter variables and backpropagated gradients can be approximated away.

[8]Real values: En 5.5, Fr 5.2, Ru 6.4.

| Method | Citation | Compress? | Generate? | Learnt? | Word level? |
|--------|----------|-----------|-----------|---------|-------------|
| GPT | Radford et al. (2018) | Lookup | Yes | No | Yes |
| ByT5 | Xue et al. (2022) | None | Yes | No | No |
| MANTa | Godey et al. (2022) | Segment | Yes | Yes | No |
| RetVec | Bursztein et al. (2023) | Conv. | No | Yes | Yes |
| FastText | Bojanowski et al. (2017) | Conv. | No | Yes | Yes |
| ELMo | Peters et al. (2018) | Conv. | No | Yes | Yes |
| CharBERT | El Boukkouri et al. (2020) | Conv. | No | Yes | Yes |
| CharFormer | Tay et al. (2021) | Conv. | No | Yes | No |
| LOBEF-nCF | Sreedhar et al. (2022) | None | Yes | Yes | No |
| LOBEF-WSF | Sreedhar et al. (2022) | None | Yes | Yes | Yes |
| CANINE | Clark et al. (2022) | Conv. | Yes | Yes | No |
| MegaByte | Yu et al. (2023) | Dense | Yes | Yes | No |
| Ours | | Attn. | Yes | Yes | Yes |

Table 7: Literature Review of existing tokenization methods along several dimensions. **Compress?** Is the input string chunked into bigger units? **Generate?** Whether or not the model can generate new unseen tokens? **Learnt?** Is the tokenization learnt end-to-end with other parameters? **Word Boundary?** Is the word boundary considered or treated as just another token? Conv: Convolution. Attn: Attention.

Subword models benefit from having tokens that are longer (roughly $2.8$ characters/subword for the three languages we consider), therefore they can generate at a speed of $2.8/t$ characters per second.

With a very coarse assumption, our end-to-end character model with $L/4$ word-encoder layers and $L$ decoder layers (ignore the $L/4$ word-decoder layers for now) will require $5t/4$ seconds to generate the representation of one word at a time. The next step can then be parallelized (with trade-off in memory consumption) to both autoregressively go on generating the next word representation in another $5t/4$ seconds, as well as autoregressively generating one character at a time using this predicted word representation. This word-level decoder that emits characters currently has $L/4$ layers so a crude assumption would mean $t/4$ seconds per character. Therefore, at steady state, the word-decoder will take $5.5t/4$ seconds to generate the average $5.5$ characters per word, while the next word will be ready for decoding simultaneously in just $5t/4$ seconds. Thus, the generation speed is $4/t$ characters per second, i.e. roughly 50% better than subwords and four times as fast as tokenizer-free models.

## 6 Related Work

Some recent work has challenged the subword tokenization schemes. Table 7 highlights the different kinds of tokenizations existing in prior work and positions our work uniquely among them.

**Character/Byte-level** ByT5 (Xue et al., 2022), CANINE (Clark et al., 2022), and SubChar (Si et al., 2021) propose using very small fixed-length units such as characters, bytes, or glyph strokes instead of dynamic-length subwords or words. This often comes at the expense of larger sequence lengths and more compute requirements, especially for a transformer architecture which typically has a complexity of $\mathcal{O}(n^2)$ in number of input tokens.

**Beyond word level** CodeBPE (Chirkova and Troshin, 2022) and Multi Word Expressions (Kumar and Thawani, 2022; Zaninello and Birch, 2020; Rikters and Bojar, 2017) show promise in yet larger tokens that cross word boundaries, e.g., a vocabulary with single tokens for the strings "for i in range" or "New York City" respectively.

**Visual segmentation** Yet another line of work (Rust et al., 2022; Salesky et al., 2021) renders text as images before feeding them to CNNs, doing away with tokenization altogether and showing gains in robustness to spelling or printing errors.

**Learnt subword segmentation** Finally, some methods (Mofijul Islam et al., 2022; Kaushal and Mahowald, 2022; Pinter et al., 2021; Tay et al., 2021; Provilkov et al., 2020; Wang et al., 2021) parameterize the process of tokenization by pooling character n-grams or randomly choosing one of the many ways to segment a given word.

A recent preprint on machine translation by Sreedhar et al. (2022) proposes a method called WSF, perhaps closest to ours, except that they only use the word boundary fusion at encoder stage. Our independent analysis focuses on language modeling instead and also generates text in parallel using end-to-end attention based tokenization.

## 7 Conclusion

Subword tokenization is efficient but too rigid and deterministic. Character/Byte-level models on the other hand are too expressive, which leads to inefficient training and inference. We propose a word-boundary-informed tokenizer that efficiently and robustly performs language modeling in a hierarchical, end-to-end, learned model. We show that it outperforms by over $300\%$ both extremes: subwords and character/byte models. We also analyze its trade-offs in training and inference efficiency. Despite its many flaws including reliance on a word boundary signal and moderate efficiency as well as moderate expressiveness, we expect this preliminary study to pose an interesting trade-off tokenization for truly end-to-end language modeling.

Our code is released on Github.

## 8 Limitations

We repeatedly highlight our work's limitations throughout the paper: our proposed end-to-end tokenization is neither faster than subwords nor as expressive as character/byte-level models. Instead, we propose it as a reasonable trade-off between the two extremes.

We also do not cast a wide net on many downstream tasks like machine translation or question answering, hence lack a comparison with other models like CharFormer, CANINE, and Local Byte Fusion. We instead focus solely on the intrinsic metrics of language modeling, across multiple languages as well as on number estimation.

Our choice of languages is limited by availability of high quality 'natural language' corpora, unlike the internet-scale language modeling data which we observed are filled with examples of long strings like URLs and unsegmented strings. We do not use preprocessing pipelines (except simple punctuation cleanup) to avoid confounding results with the choice of such heuristics. This unfortunately prevents us from experimenting with other high resource languages like Chinese and Japanese, corpora of which do not implicitly have a whitespace

boundary signal.

To summarize, we concede that our proposed end-to-end models are not yet ready for adoption at scale with large language models trained on raw internet data. However, we expect our analyses to encourage insightful conversation in the community about (1) the spectrum between subwords and character/byte models, as well as on (2) the role of word boundary as a meaningful signal in tokenization and language modeling.

## 9 Acknowledgements

This work was funded by the Defense Advanced Research Projects Agency with award N660011924033. The authors acknowledge the Center for Advanced Research Computing (CARC) at the University of Southern California for providing computing resources that have contributed to the research results reported within this publication. We also thank Google for their generous support. We appreciate the anonymous reviewers at EMNLP 2023 for helping us refine this paper.

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
