# OpenReview forum: "Learn Your Tokens: Word-Pooled Tokenization for Language Modeling"
_EMNLP/2023/Conference — EMNLP 2023 Findings_

### Official Review · Reviewer_nfbm · 2023-07-27

**Soundness:** 4

**Excitement:**

4: Strong: This paper deepens the understanding of some phenomenon or lowers the barriers to an existing research direction.

**Missing References:**

None found.

**Paper Topic And Main Contributions:**

This paper presents a new tokenization technique for language modeling, which is an alternative to both subword tokenization and raw character or byte-based tokenization. The proposed method learns tokens by using word boundaries to pool characters or bytes into a word representation (in particular, the character or byte representations of each word are compresed into a constant number n of CLS tokens, the paper tries n=1 and n=4). This scheme has better efficiency than byte/char tokenization, albeit worse than subword tokenization (because attention is computed between bytes within the same word, but not those in different words), and is also a middle ground in terms of expressivity. The most salient result is language modeling accuracy, with the proposed scheme outperforming both subword and byte/char-based tokenization by a large margin, and even more so on rare words (by a factor of 30 in this case).

**Questions For The Authors:**

n/a

**Reasons To Accept:**

The paper addresses a very relevant and timely subject, since tokenization is the first step of any NLP task and in particular the first step before applying the now ubiquitous LLMs. The paper makes a good point that LLMs learn tasks almost completely end-to-end, except for tokenization, and gives good explanations of the problems this causes.

The paper is clearly written, easy to follow, and the experimental methodology is sound.

The results are very good. The proposed method strikes a balance between efficiency and expressivity, while being very good (much better than the subword and char/byte-based alternatives) at next word prediction.

Experiments are rather wide in terms of covering various languages, and doing specific experiments about arithmetic, a weak point of other tokenization schemes.

The paper is very sincere and clear about the limitations of the approach.

**Reasons To Reject:**

Experiments are all done on a single autoregressive language model, which is relatively small, and there is no guarantee that the results will generalize to other language models or that they will scale. I don't consider this to be a real reason to reject as not everyone has the resources to try other/larger models.


**Reproducibility:**

4: Could mostly reproduce the results, but there may be some variation because of sample variance or minor variations in their interpretation of the protocol or method.

**Reviewer Confidence:**

3: Pretty sure, but there's a chance I missed something. Although I have a good feel for this area in general, I did not carefully check the paper's details, e.g., the math, experimental design, or novelty.

**Typos Grammar Style And Presentation Improvements:**

Why is eChar missing from Table 6?

Line 24: outperform -> outperforms

Line 256: pipleine -> pipeline

---

> ### Author Rebuttal · Authors · 2023-08-29
>
> Thank you for taking the time to thoroughly review our paper and provide insightful comments!
>
> We specifically appreciate your consideration of limited resources when evaluating the limited scale of our experiments and instead judging our work on the soundness and novelty of the research questions considered.
>
> Here are the eChar and Char results as requested for Table 6:
> - eChar: 46.7 (% Num), 45.6 (EAcc), 90.0 (MdAPE).
> - Char: 42.8 (% Num), 46.6 (EAcc), 92.5 (MdAPE).
>
> We will be happy to include them in the camera ready version.

---

### Official Review · Reviewer_kEqs · 2023-08-02

**Soundness:** 3

**Excitement:**

4: Strong: This paper deepens the understanding of some phenomenon or lowers the barriers to an existing research direction.

**Paper Topic And Main Contributions:**

This paper uses very small text dataset and make a statement on what kind of tokenizer that is more suited for language modelling. The authors propose to pool word representations into the LM and decoding them at the character/byte level. The method is said to improve word accuracy across dataset and works particularly well on rare words.

**Questions For The Authors:**

some corrections:

What is word accuracy? please explain your metric. Also is it used in other papers? if yes please add citations.

Your improvement is not exactly 300% so either say it is 314% or just that is shows a 3 fold increase.

line :whereas some African scripts may require four bytes to represent a single character [citation  missing, or at least an example of such language]

line 529: We do not using

line 298: we report magnitude-based metrics that are typically reported in the literature [missing reference]. Also what is EAcc and MdAPE ? please explain briefly

**Reasons To Accept:**

The paper is clearly written and simple to understand.

**Reasons To Reject:**

The model used is typically a megabyte with dynamic size tokens that respect the word boundaries. Even though the idea is interesting, the paper is not using it in a convincing manner. Your model is trained to decode the real words for many epochs (100) on a very small dataset. Such a model is expected to have learned a good embedding for all words in the corpus, even for rare words. It is not surprising that such a model will outperform subword and character based model that have not been trained to decode exactly the complete words. As this paper does not prove that the method is of any use in real word scenario (machine translation, sentence generation,...), it is difficult to understand the take away from this paper.

I think this paper is too weak to be accepted at EMNLP. Yet, an idea that would be interesting to pursue would be to compare your model to Megabytes that always pool 4 characters without taking into account the word boundaries. If you manage to show that dynamic size tokens are better than fixed-size ones, you would have an interesting results to publish.

**Reproducibility:**

5: Could easily reproduce the results.

**Reviewer Confidence:**

4: Quite sure. I tried to check the important points carefully. It's unlikely, though conceivable, that I missed something that should affect my ratings.

---

> ### Author Rebuttal · Authors · 2023-08-29
>
> Thank you for your insightful comments and interesting ideas for follow-up experiments!
>
> As you mention, it will be interesting to compare our results with MegaByte’s fixed-length compression, which we are already working on. We would nevertheless stress that it was unrealistic to include such results in the current submission, given that **[MegaByte](https://arxiv.org/abs/2305.07185) is an unpublished preprint submitted to arxiv less than 5 weeks before the EMNLP deadline**. We included MegaByte in our Related Work section merely to taxonomize the literature review and to highlight the vast interest shown by the NLP community in the search for alternative end-to-end tokenization strategies.
>
> We will be happy to further describe our metrics like word prediction and numeracy metrics described in L300 and Table 6: “EAcc = Exponent Accuracy. MdAPE = Median Absolute Percentage Error.” Word prediction evaluates whether the next characters/bytes/subwords predicted include the exact next ground truth word. Exponent Accuracy is an order-of-magnitude accuracy (whether the number of digits are the same in the ground truth and predicted number). Median Absolute Percentage Error is formulated as the median of 100|x-y|/y where x is the prediction and y is the ground truth number. We will cite relevant prior work for these metrics:
> - Spokoyny, Daniel, and Taylor Berg-Kirkpatrick. "[An empirical investigation of contextualized number prediction](https://aclanthology.org/2020.emnlp-main.385/)."
> - Thawani, Avijit, Jay Pujara, and Filip Ilievski. "[Numeracy enhances the literacy of language models](https://aclanthology.org/2021.emnlp-main.557/)."
>
> We will also make sure to correct other minor typos and paraphrasing errors that you point out.
> We will also make sure to correct other minor typos and paraphrasing errors that you point out. Bamum is an example of an African script that suffers from using many bytes merely because it is placed far in the [Basic Multilingual Plane](https://unicode.org/roadmaps/bmp/). Other disadvantaged scripts are Meetei (India) and Cherokee (North America).
>
> Finally, we would like to challenge the assertion that our proposed end-to-end models are unfairly trained on word prediction. All model variants were trained for the same number of epochs and the number of epochs was determined by looking at validation loss curves (which we could add to the camera ready appendix for justification). The training proceeds with negative log likelihood loss over the base units: characters for eChar and Char; bytes for eByte and Byte; and subwords for subword model. There is no added signal that unfairly helps our proposed end-to-end tokenized models excel at the task of word prediction.
>
> We acknowledge (as in our Limitations section) that this paper does not convincingly conclude which tokenizer must be used for training language models in general. Our datasets are too simple and small, and we do not cast a wide net on downstream tasks. Nevertheless, our hope is that **our systematic analysis of tokenizers will benefit the community**, given how difficult it is to compare such variants due to the need for pretraining language models from scratch.
>
> We hope we have sufficiently answered your queries about the soundness of our approach. Regarding the excitement/novelty of our approach, we *kindly request that you compare our work against only non-contemporaneous work* (published over three months before the EMNLP deadline).

---

### Official Review · Reviewer_Exn2 · 2023-08-04

**Soundness:** 3

**Excitement:**

3: Ambivalent: It has merits (e.g., it reports state-of-the-art results, the idea is nice), but there are key weaknesses (e.g., it describes incremental work), and it can significantly benefit from another round of revision. However, I won't object to accepting it if my co-reviewers champion it.

**Paper Topic And Main Contributions:**

This work proposes a new tokenization scheme for language modeling. This scheme uses a Transformer encoder to pool in the base tokens (either characters or bytes) into a fixed number of embeddings per word. Those embeddings are, then, passed onto the main language model (in this case, a vanilla GPT-like Transformer decoder) and, after that, fed trough a Transformer decoder to autoregressively decode the next word.

The proposal is evaluated in comparison to other well-known strategies (subword-level, byte-level and character-level) over language modeling, showcasing significant improvements in terms of accuracy (with an increase of memory consumption as a trade-off). However, I'm not convinced that the evaluation has been done under the same conditions for all strategies (e.g., Section 3.1 mentions that the subword strategy is the BPE vocabulary from GPT-2 and GPT-3) nor I am sure that the language model implementation is the same for all strategies (though this may just be a problem with the description in Section 3.4).

As mentioned at the limitations section, both the scalability to broader data and the use of other languages are a main concern.

**Questions For The Authors:**

As I'm understanding Section 3.4, when you use expressions such as "Our base language model is", it makes me think that your are using a different architecture for your language model, instead of using the same one replacing only the tokenization strategy. Is that so?

**Reasons To Accept:**

* Novel strategy that could benefit the language model community.

**Reasons To Reject:**

* We are uncertain of the scalability to broader data and languages.

**Reproducibility:**

3: Could reproduce the results with some difficulty. The settings of parameters are underspecified or subjectively determined; the training/evaluation data are not widely available.

**Reviewer Confidence:**

4: Quite sure. I tried to check the important points carefully. It's unlikely, though conceivable, that I missed something that should affect my ratings.

**Typos Grammar Style And Presentation Improvements:**

l. 086 and 087: (Tay et al., 2021; Yu et al., 2023; Clark et al., 2022) -> (Tay et al., 2021; Clark et al., 2022; Yu et al., 2023)
l. 227: byte level -> byte-level
l. 228: Xue et al. (2022) -> (Xue et al., 2022)
l. 250: "192//N" is not standard notation
l. 312 and 319: use em-dash instead of dash
Table 6: dastaset -> dataset
l. 530: using -> use

---

> ### Author Rebuttal · Authors · 2023-08-29
>
> Thank you for your thoughtful comments! We appreciate your feedback that Section 3.4 should be reworded for more clarity. In the meanwhile, we want to emphasize that **evaluation was indeed done under the same conditions for all tokenizers**. Our reference to “the base language model” refers to the underlying transformer decoder architecture, which remains the same for all strategies (all different tokenizers).
>
> We understand the concern around scalability and generalizability of our results. Despite our findings being consistently across multiple languages and multiple metrics, we note that it is natural for a new tokenization method to undergo years of empirical experimentation across tasks, models, and languages before wide adoption, e.g., the now-ubiquitous subwords were once introduced in papers working exclusively on machine translation on a few language pairs. This paper is, therefore, a preliminary work to very specifically study the difference in language modeling capabilities across several tokenization strategies.
>
> We hope we have clarified the concerns around the soundness of our work by reiterating the fair experimental setup across experiments. Regarding excitement around the scalability and generalizability of our results, we request that you consider the **resources required to compare tokenization in language models by pretraining each variant from scratch**. Given a modest computation budget, we have tried our best to contribute insightful results across tokenizers for the community to learn from, and scale up our experiments across models and languages in future work.

---

### Meta-Review · Area_Chair_YLV6 · 2023-09-25

**Recommendation:** 3

**Metareview:**

Authors propose a new tokenization method for language modeling alternative to other sub-word tokenization or byte/char-level methods. Reviewers all acknowledged the merits of the proposed approach but also highlighted the need for a more controlled setup when it comes to the granularity of the learning signal and how it compares with the other methods, since this could be a confounder needs studying and further explanations.

---

### Decision · Program_Chairs · 2023-10-07

**Decision:**

Accept-Findings

**Comment:**

Authors propose a new tokenization method for language modeling alternative to other sub-word tokenization or byte/char-level methods. Reviewers all acknowledged the merits of the proposed approach but also highlighted the need for a more controlled setup when it comes to the granularity of the learning signal and how it compares with the other methods, since this could be a confounder needs studying and further explanations.